# Advancing ASL Kidney Image Registration: A Tailored Pipeline with Groupwise VoxelMorph

**Anne Oyarzun-Domeño**[1,2]                                    ANNE.OYARZUN@UNAVARRA.ES
[1] *Electrical Electronics and Communications Engineering, Public University of Navarre, 31006, Pamplona, Spain.*
[2] *IdiSNA, Health Research Institute of Navarra, 31008, Spain.*

**Izaskun Cia**[1]                                              IZASKUN.CIA@UNAVARRA.ES
**Rebeca Echeverria-Chasco**[2,3]                              RECHEVERRIAC@UNAV.ES
[3] *Department of Radiology, Clínica Universidad de Navarra, 31008, Pamplona, Spain.*

**María A. Fernández-Seara**[2,3]                              MFSEARA@UNAV.ES
**Paloma L. Martin-Moreno**[2,4]                              PLMARTIN@UNAV.ES
[4] *Department of Nephrology, Clínica Universidad de Navarra, 31008, Pamplona, Spain.*

**Nuria Garcia-Fernandez**[2,4]                               NRGARCIA@UNAV.ES
**Gorka Bastarrika**[2,3]                                      BASTARRIKA@UNAV.ES
**Javier Navallas**[1,2]                                       JAVIER.NAVALLAS@UNAVARRA.ES
**Arantxa Villanueva**[1,2,5]                                  AVILLA@UNAVARRA.ES
[5] *Institute of Smart Cities (ISC), Health Research Institute of Navarra, 31006, Pamplona, Spain.*

## Abstract

Arterial spin labeling (ASL) provides a non-invasive assessment of renal blood flow, but it faces difficulties due to motion artifacts and the effects of blood inflow. This work introduces GVox, a deep learning-based motion correction (MoCo) framework tailored for ASL imaging. GVox extends VoxelMorph, incorporating cortical signal enhancement as metric to optimize and groupwise inference as main contribution. Proposed GVox demonstrates superior performance compared to the baseline Elastix, with significantly improved image similarity and computational efficiency.

**Keywords:** Arterial spin labeling, deep learning-based image registration, cortical signal.

## 1. Introduction

ASL is a magnetic resonance imaging (MRI) technique that enables the characterization of renal blood flow by magnetically label arterial blood water spins in a non-invasive way (Nery et al., 2018). *Control* images contain static tissue signal while *label* images contain both static tissue signal and tagged blood signal. Perfusion (or ASL) maps are computed by substracting control and label images voxel by voxel. Added to the motion caused by the anatomical displacement of the kidney (due to respiratory and motion of the patient) is the presence of blood inflow that expands through the kidney, that being the case of misregistered image sequences and subsequent post-processing errors. Recent deep learning-based image registration (DLIR) approaches use convolutional neural networks (CNN) for the estimation of voxel or pixel-wise spatial correspondences (Zöllner et al., 2009), such as VoxelMorph (Balakrishnan et al., 2019). DLIR approaches model the function $g_\theta(I_f, I_m) = \phi$. $I_f$ and $I_m$ are the fixed and moving images over a $n$-D spatial domain $\Omega \, \epsilon \, \mathbb{R}^n$, respectively, $\phi$ is the deformation field, and $\theta$ are the learnable parameters of function $g$ that compute

the optimal deformation field $\phi$ for each image pair $(I_f, I_m)$. The model warps $I_m(p)$ into $I_m(\phi(p))$ by spatial transformer network (STN) (Jaderberg et al., 2015) based module, considering $p$ as a pixel in a 2D image. The spatial transformation enables the calculation of dissimilarity loss function named as $F_{sim}$ and updates $\theta$.

**Aim**. We propose the following pipeline for ASL MoCo on healthy kidneys: 1) to extend VoxelMorph to provide a tool tailored to the unique needs of ASL, 2) to introduce a cortical enhancement based measurement as a customized loss metric used as a complement to the main similarity loss function, and 3) to implement groupwise inference stage by using principal component analysis (PCA).

## 2. Method

**Data**. The dataset consists of 48 healthy kidney ASL studies with acquisition matrix of 96 x 96. Each study contains a $M_0$ proton density or reference image and a set (8-25 pairs) of control/labels ASL pairs. Manually drawn cortical masks are used for training. Images are intensity-normalized $\{0-1\}$ and affinely registered with Elastix before training.

**Registration pipeline** Our MoCo framework consists of separate training and testing stages (Figure 1a). *i. Training*. MoCo models are trained using VoxelMorph-2 architecture (Balakrishnan et al., 2019) in unique study (higher motion measured), Adam optimizer, learning rate of $10^{-4}$, 1000 epochs, 100 steps per epoch, a batch size of 8, and weighting regularization parameter ($\lambda$) of 0.9 (experimentally set). The method is implemented on Python 3.7 using PyTorch 1.13 on GPU NVIDIA GeForce RTX 3060. Our models are trained using both unidirectional and bidirectional registration (Figure 1a). We use a customized loss function that measures $F_{sim}$ between randomly selected $I_f$ and $I_m$ pairs, consisting of a weighted sum of normalized cross correlation ($NCC$) and cortical temporal signal-to-noise ratio ($tSNR$) (Equation 1). $tSNR$ is calculated as the ratio of time series (random input images on training) average perfusion signal and its temporal standard deviation (SD), over the manually drawn ROI of the cortex.

$$F_{sim}(I_f, I_w, S_{I_f}, S_{I_w}) = -\omega_{ncc}NCC(I_f, I_w) - \omega_{tsnr}tSNR((S_{I_f}, S_{I_w})) \qquad (1)$$

where $I_w$ is $I_m \circ \phi$, $S_{I_f}$ and $S_{I_w}$ are the cortical masks corresponding to $I_f$ and $I_w$, being $S_{I_w} = S_{I_m} \circ \phi$. $\omega_{ncc}$ and $\omega_{tsnr}$ are the corresponding weights for $NCC$ and $tSNR$, set as 0.9 and 0.1, respectively. *ii. Testing*. Intra-patient groupwise MoCo is achieved by evaluating trained models on unseen studies by iteratively warping $N$ images into PCA-based template image $T$. The eigenvector $V = (w_1, w_2, ...w_n)$ associated with the largest eigenvalue is used as weights for the construction of $T$. In each iteration, the process randomly designates a moving image and registers it with the template image $T$. $T$ is initialized using $N$-1 non-warped images from the unregistered sequence. As the inference advances, images from the unregistered sequence are systematically updated with warped, so it does $T$.

**Baseline**. We use Elastix (Klein et al., 2010) as the reference non-learning-based MoCo method due to its widespread use in ASL MoCo (Nery et al., 2019; Bones et al., 2019).

## 3. Results

We use mean structural similarity index (MSSIM) for MoCo evaluation, that assesses the structural similarity of images by comparing luminance, contrast, and structure (Sassi et al.,

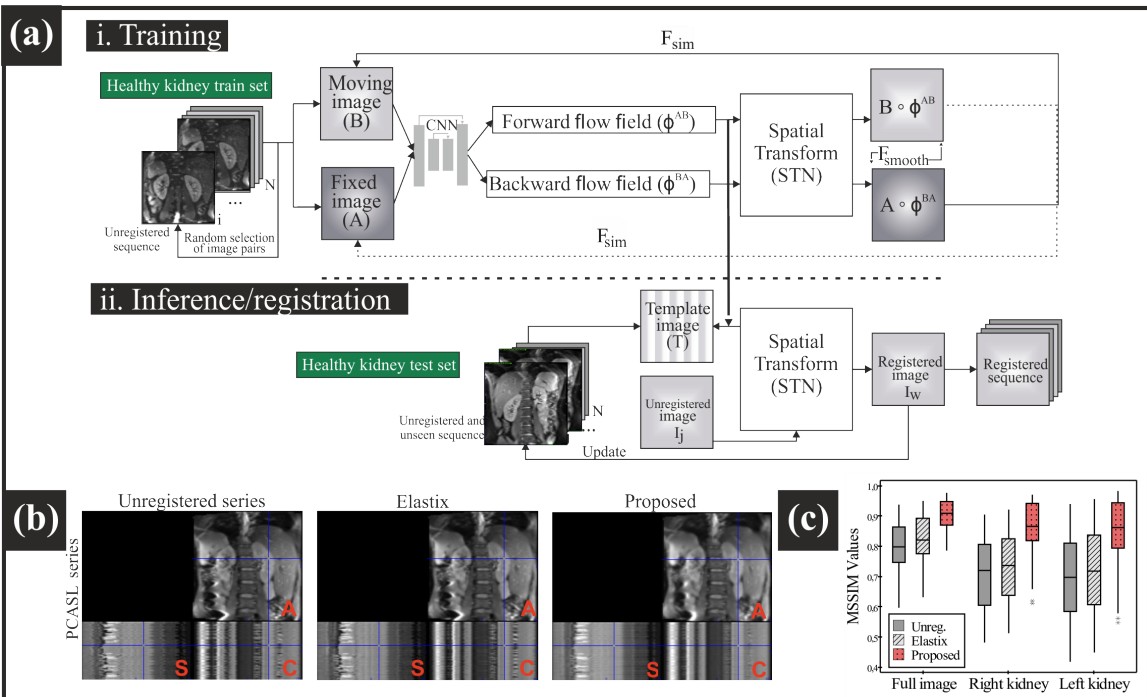

Figure 1: Overview of GVox. a) MoCo pipeline. b) MoCo effects on image alignment (axial (A), sagittal (S), and coronal (C) views). c) Boxplot showing MSSIM values.

2008). MSSIM is calculated in the whole image, and in the right and left kidney encompassing bounding box, separately. For statistical analysis, we use ANOVA with Dunnet pairwise comparison, where unregistered series is set as the control group. MSSIM was significantly higher ($p<0.05$) in GVox compared to both unregistered series and Elastix. Regarding the evaluation of registration direction, none statistical ($p > 0.05$) difference was found in MSSIM metric between unidirectional and bidirectional training frameworks. Mean registration times for Elastix and GVox are 76 and 9 secs per epoch, respectively. Thus, GVox demonstrates higher computational efficiency. Qualitative results (Figure 1b) demonstrate excellent performance of GVox in sequentially aligning images.

## 4. Conclusions

Our GVox method shows significant promise in renal ASL MoCo, preserving anatomical structure while delivering efficient runtimes and superior performance in image similarity. These findings suggest the potential of GVox to enhance the accuracy and reliability of medical MoCo for renal imaging. Moreover, groupwise registration allows for the inclusion of full ASL data series. As future work, our method shows potential for enhancing renal imaging accuracy and reliability, particularly in challenging scenarios such as transplant patients and chronic kidney disease patients with inconsistent breathing patterns.

## Acknowledgments

Project PC181-182 RM-RENAL and predoctoral grant 0011-0537-2021-000050, by the Department of University, Innovation and Digital Transformation, Government of Navarra.

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
