# OpenReview forum: "Advancing ASL Kidney Image Registration: A Tailored Pipeline with Groupwise VoxelMorph"
_MIDL.io/2024/Short_Papers — MIDL 2024 Short Papers_

### Official Review · Reviewer_9mou · 2024-04-24

**Confidence:** 5
**Final Rating:** 3.5

**Review:**

This paper presents GVox, an application of a well-known deep learning-based nonlinear registration method, VoxelMorph, in a groupwise matter for the motion correction problem in kidney ASL imaging data. It is a straightforward application rather than presenting a novel idea. Moreover, it was compared only with Elastix (which is not a very old nonlinear registration method).

---

### Decision · Program_Chairs · 2024-04-26

Accept